# GraphMind: LLMs as Dynamic Knowledge Builders for Sequential Decision-Making

## Abstract

While the reasoning capabilities of large language models (LLMs) have advanced considerably due to their extensive internal knowledge, efficiently internalizing and leveraging new information in dynamic environments remains as a significant challenge. This limitation is particularly pronounced in partially observable environments, which require agents to manage long-term memory and perform effective exploration under incomplete information. To address this, we propose an LLM agent architecture that integrates a knowledge graph as a graph-based memory module to facilitate high-level action planning. The agent incrementally constructs the knowledge graph through environmental interactions and retrieves relevant information to generate efficient plans. We evaluate our approach in complex navigation tasks specifically designed to present long-horizon and partially observable challenges. Experimental results demonstrate that incorporating a knowledge graph as an external memory significantly enhances the success rate and efficiency of the LLM's planning capabilities.

## 1 Introduction

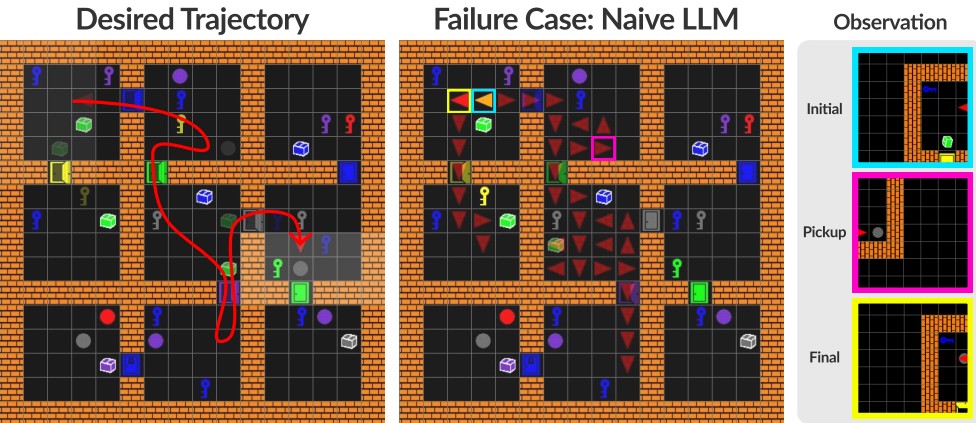

Figure 1: Examples of a desired trajectory (**left**) and a failed trajectory (**middle**) in the partially observable mission "put a gray ball next to the green key." The agent moves from the start position (▶) to the final position (▶). The naive LLM planner fails to complete the mission due to insufficient exploration.

Large language models (LLMs) have demonstrated remarkable performance in natural language understanding and generation, establishing themselves as foundational tools across a wide range of domains. Recently, research has increasingly focused on leveraging LLMs for interaction with dynamic environments, exploiting their strong prior knowledge and reasoning capabilities. Studies such as Carta et al. (2023); Paglieri et al. (2024) have reported promising results in sequential decision-making tasks, highlighting the potential of LLMs as agents in complex, interactive settings. These approaches typically rely on a short context window over recent trajectories, limiting the agent's ability to retain and exploit long-term context during decision-making.

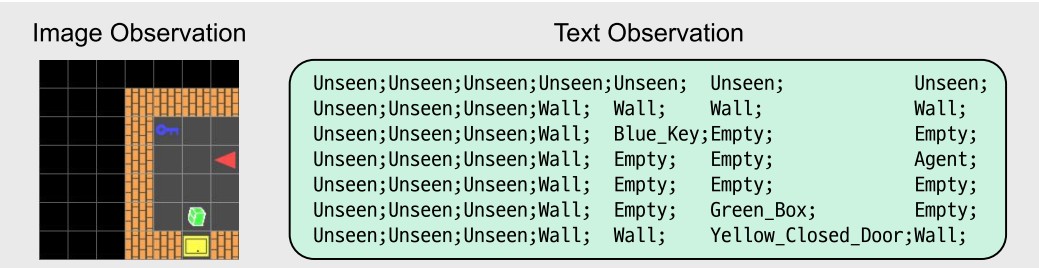

Figure 2: Two observation formats: a pixel-based image and a textual description.

However, many real-world tasks are inherently long-horizon and complex, and the additional challenge of partial observability requires agents to maintain and reason over extended context in order to make effective decisions. Figure 1 illustrates an example of an environment that presents substantial difficulties for a naive LLM planner, particularly under partial observability. In standard natural language processing tasks such as question answering or text generation, Retrieval-Augmented Generation approaches (Lewis et al., 2020; Yu et al., 2022; Han et al., 2024) address this issue by retrieving relevant chunks from large-scale external documentation. Yet, these methods are constrained by the static nature of the external knowledge sources: the documentation is fixed and cannot account for dynamically expanding information generated through ongoing interaction with an environment. This limitation motivates the development of new mechanisms for adaptive memory construction and retrieval tailored to sequential decision-making under partial observability.

Another line of work explores expanding external memory to support long-horizon tasks. For instance, studies by Anthropic (2025) and Comanici et al. (2025) utilize the challenging benchmark of Pokémon Red to evaluate long-term memory in LLMs. Instead of persistently including all information in the prompt, both approaches equip the agent with tools for on-demand knowledge retrieval, enabling a form of extended reasoning that streamlines the decision-making process. Specifically, the approach by Anthropic (2025) with Claude Opus 4 maintains external memory files to store key information. Similarly, the approach by Comanici et al. (2025) with Gemini-2.5-Pro condenses action sequences in batches to reduce input tokens. This summarization focuses on event sequences rather than constructing a spatial mental map. While maintaining continuity, both approaches result in an inefficient memory structure and substantial storage requirements, limiting scalability.

To address these limitations, we propose *GraphMind* [1], a scalable and effective self-expanding external memory framework with two key components. First, inspired by prior work on knowledge graphs Pujara et al. (2013), we organize information from past interactions into a graph-based representation. This provides a compact yet expressive memory mechanism, particularly advantageous in partially observable navigation tasks, as it explicitly captures spatial relationships between objects and locations. Second, to support efficient decision-making, we augment the planning capabilities of LLMs with a domain-specific language to enable structured reasoning and planning. This combination improves exploration efficiency, which is critical under partial observability, while grounding the knowledge graph in trajectories collected through the actor's behavior. Our experiments demonstrate that the proposed structured approach enables an LLM agent to complete tasks in challenging long-horizon, partially observable environments.

## 2 DOMAIN AND PROBLEM STATEMENT

Complex, long-horizon tasks are common in real-world settings and require both effective memory mechanisms and advanced planning capabilities (Hu et al., 2025; He et al., 2025). Among alternatives such as Blocks World (McDermott, 2000), we adopt and extend BabyAI (Chevalier-Boisvert et al., 2019), a partially observable 2D gridworld that combines diverse challenges: object manipulation, navigation, exploration–exploitation trade-offs, and mission execution specified in the simplified text-based Baby Language. In contrast to other environments such as ALFWorld (Shridhar et al., 2020) and MiniHack (Samvelyan et al., 2021), BabyAI provides an Oracle Solver (referred to as BOT in (Chevalier-Boisvert et al., 2019)), which generates step-by-step solutions using hand-coded

---

[1]Our code is available at: https://anonymous.4open.science/r/GraphMind-1080.

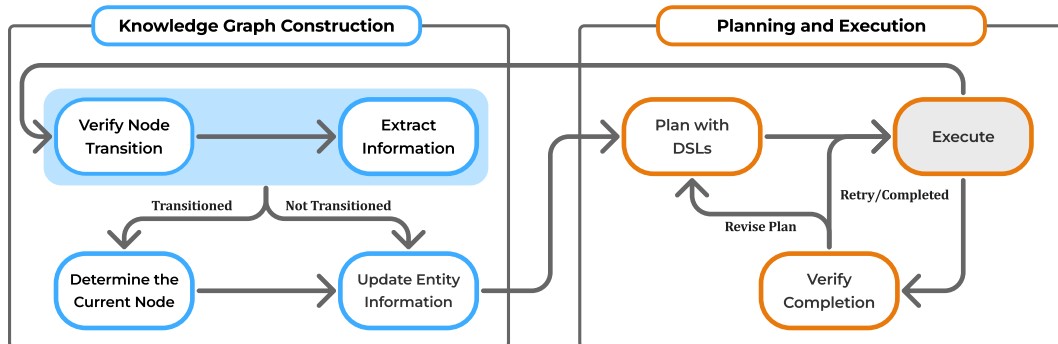

Figure 3: Overview of the proposed framework. The left panel illustrates the construction of the knowledge graph, where the agent verifies node transitions, extracts information from observations, determines the current node, and updates entity information. The right panel depicts the planning and execution loop, in which the LLM generates DSL-based plans, the actor executes instructions, completion is verified, and plans are revised if necessary. Arrows indicate the flow of information and iterative feedback between modules.

rules and an internal stack machine. This feature enables systemic evaluation of how our method expands memory and influences planning in long-horizon, partially observable environments.

Each environment layout consists of $n$ rooms connected by colored doors and populated with color-coded objects. The agent explores these rooms to locate target objects, while its field of view is restricted by walls and doors, as shown in Figure 2. Unlocked doors permit free traversal, whereas locked doors require keys of the corresponding color, thereby increasing the demand for exploration. To further emphasize partial observability, we modified the environment such that open doors also block the line of sight, preventing perception beyond the doorway. The detailed discussion on the multimodal observations is provided in the appendix C.

In our experimental setup, we focus on two challenging missions, `OpenDoor` and `PutNextTo`, that have proven difficult in prior work (Carta et al., 2023). In `OpenDoor`, the agent must retrieve a key and unlock a corresponding door located elsewhere in the layout. To guarantee that tasks require exploration, we randomly generate diverse layouts and missions, filtering them to ensure that key entities are placed in non-adjacent rooms. In `PutNextTo`, the agent must find two distinct objects placed in separate rooms and bring them together, making success contingent on navigating multiple rooms rather than exploiting local information. Additional details are provided in Appendix B.

## 3 PROPOSED APPROACH

We propose *GraphMind*, a framework that enables agents to operate effectively in partially observable environments. GraphMind dynamically constructs a self-expanding graph-based memory from collected observations and employs adaptive planning to retrieve and exploit the knowledge required for task completion. Graph structures provide a compact and expressive way to encode large-scale, heterogeneous, and relational information, in contrast to linear structures. These structures are particularly well-suited for capturing spatial relationships, which are critical in navigation tasks (Hogan et al., 2021; Han et al., 2024).

The proposed framework operates in two iterative stages: (1) **Knowledge Graph Construction**, where observations are integrated into an incrementally expanding graph representation, and (2) **Planning and Execution**, where a domain-specific language (DSL) supports structured reasoning and action selection to gather task-relevant information under partial observability. This iterative process enables systematic expansion of the agent's knowledge base and progressively improves task performance. An overview of our approach is illustrated in Figure 3, highlighting the interaction between graph construction and planning. Detailed descriptions of each module and the corresponding prompts are provided in the Appendix A. An overview of our approach is illustrated in Figure 3, highlighting the interaction between graph construction and planning. Detailed descriptions of each module and the corresponding prompts are provided in the Appendix A.

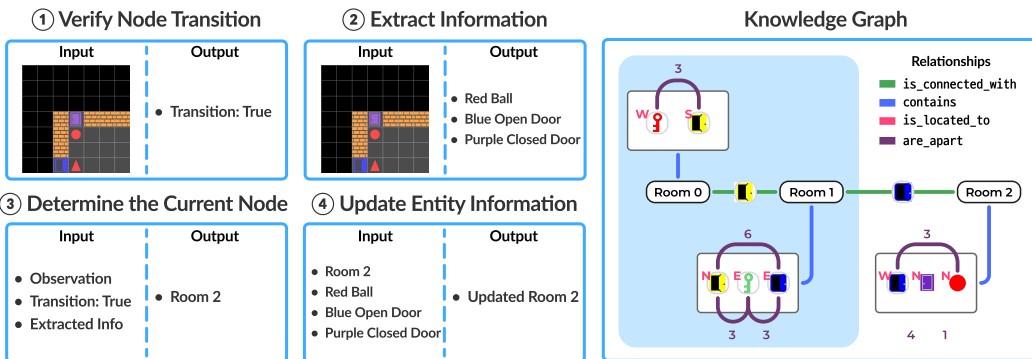

Figure 4: Example of updating the knowledge graph in BabyAI. When the agent (▶) observes a red ball (●) and a purple door (▣) after traversing a blue door (■), the corresponding modules verify the door traversal, extract object information, and expand the graph by adding a new room node, its connecting edge, and the associated relationships, building upon the existing graph (shaded area).

## 3.1 KNOWLEDGE GRAPH-BASED MEMORY

We propose a graph-based memory representation that expands dynamically with the help of LLMs. Nodes correspond to entities or locations, and edges encode relationships through relational predicates. This design allows the agent to capture both spatial and semantic information, which are essential for reasoning and planning in partially observable environments. We focus on four relation types: is_connected_with (spatial connectivity), contains (object presence within a space), is_located_to (directional attributes), and are_apart (relative distances that help distinguish otherwise identical entities). In our adapted setup, rooms are modeled as nodes, doors as edges denoting connectivity, and each room node is annotated with contained objects via contains and is_located_to. Objects within the same room are linked by are_apart, while adjacent rooms are joined by is_connected_with. Figure 4 shows an example of constructing such a graph from agent observations.

Figure 3 (left) illustrates knowledge graph construction, implemented via four dedicated modules:

- **Verify Node Transition**: analyze the agent's action sequence to detect transitions and update is_connected_with relations.
- **Extract Information**: parse the current observation to identify objects, expand contains relations, and annotate entities with directional and distance attributes to establish is_located_to and are_apart.
- **Determine Current Node**: localize the agent by integrating observed entities, verified transitions, and connections to the previously known location.
- **Update Entity Information**: merge extracted attributes into the current node to maintain a coherent, up-to-date representation of the environment.

The graph expands incrementally as the agent explores, with LLM-based modules guiding construction. To improve robustness, we apply an ensemble method to refine is_connected_with inference. This process yields an adaptive memory structure that evolves continuously during interaction, supporting long-horizon reasoning.

**External Tools for Knowledge Utilization** We augment the LLM with external tools to retrieve relevant information from a knowledge graph, enabling effective utilization of structured environmental knowledge. Inspired by prior works that leverage tool use for efficient prompting (Anthropic, 2025; Comanici et al., 2025), the proposed method similarly avoids including the full summarized memory in every prompt. Instead, the LLM determines whether to request the information based on its current observation, enabling the agent to retrieve from external memory. Given a room identifier, the tools return detailed information about the specified room, including its adjacent rooms and the entities contained therein. Additionally, the shortest path from a given room to the nearest target entity is computed using the breadth-first search algorithm. The integration of the knowledge graph with external memory tools is motivated by its efficiency in managing information in partially observable environments. In such settings, the memory requirements are dynamic, expanding as the agent explores, which poses significant challenges for scalable and effective memory management.

Table 1: DSLs and their descriptions for BabyAI environment.

| DSLs | Description |
|---|---|
| find_door() | Find a door in the current room. |
| pass_door(x) | Pass through door x. |
| go_to_entity(x) | Move to face entity x. |
| pick_up_entity(x) | Pick up entity x. |
| drop_entity(x) | Drop the currently held entity x on an empty cell. |
| drop_next_to_entity(x,y) | Drop the currently held entity x next to entity y. |

## 3.2 PLANNING USING DOMAIN-SPECIFIC LANGUAGE

A fundamental challenge in LLM-based agents lies in bridging the gap between high-level planning and low-level action execution. LLMs excel at generating symbolic, abstract, and commonsense-driven plans but often lack the precision and reliability required for fine-grained control in dynamic environments (Ma et al., 2024; Wen et al., 2024). In contrast, low-level action policies—whether heuristic controllers or reinforcement learning agents—are effective at executing primitive behaviors but lack the ability to reason about long-term dependencies or abstract objectives. Our framework addresses this disconnect through the use of domain-specific languages (DSLs), defined as computer languages tailored to particular application domains, which provide a structured interface between the symbolic reasoning of the LLM and the concrete action space of the environment.

DSLs have been widely adopted to enhance the reasoning and problem-solving capabilities of LLMs in structured tasks (Barke et al., 2024; Chollet et al., 2024). In our framework, the DSL provides a compact yet expressive action space, enabling the agent to efficiently navigate, manipulate objects, and execute high-level strategies required for task completion. The DSL comprises six instructions, enumerated in Table 1. By encoding navigation and interaction primitives as DSL instructions, the LLM operates at the level of high-level goals while delegating execution details to a low-level actor. This separation of concerns reduces the cognitive load on the LLM and enhances robustness in action execution. Consequently, our approach effectively aligns symbolic reasoning with embodied interaction, bridging a longstanding gap between high-level planning and low-level control in partially observable environments.

We formalize the decision process as an iterative "plan–execute–verify–revise" loop, as illustrated in Figure 3 (right):

- **Plan with DSLs**: At each step, the LLM generates a sequence of DSL instructions conditioned on the current observation and retrieved knowledge graph information accessed via external tool calls.
- **Execute**: The actor executes the next instruction from this sequence.
- **Verify Completion**: After execution, the system assesses whether the intended subgoal has been achieved.
- **Revise Plan**: If verification fails, the framework either adapts the plan by prompting the LLM to generate a revised set of DSL instructions or allows the actor to retry the current instruction.

This cyclical structure ensures that planning remains adaptive, resilient to execution errors, and robust under partial observability.

## 4 EXPERIMENTS

Our experiments were conducted in complex multi-room environments, in contrast to the simple, single-room setups used in prior work (Carta et al., 2023; Paglieri et al., 2024). The environments follow grid layouts of $2 \times 2$ and $3 \times 3$ with complexity further increased by including at least one locked door in each layout. As described in Section 2, we filter layouts to ensure that accessing the target objects requires obtaining a key to unlock a door. In particular, for $3 \times 3$ layouts, we enforce that completing the mission necessitates exploring at least four rooms. We evaluated two models from the Google DeepMind Gemini 2.5 series (Comanici et al., 2025): **Gemini-2.5-Flash** and **Gemini-2.5-Pro**. To account for the non-stationarity of partially observable environments and the variability of LLM responses, we conducted three trials for each layout. To ensure environmental diversity, we generated five random layouts under two entity-density conditions: one entity

Table 2: Success rate of completed missions (`PutNextTo` and `OpenDoor`). For graph edit distance (GED), lower values indicate a more accurately constructed knowledge graph. We adopted expert bot heuristic bot as a downstream actor. We denote our framework variants as follows: with the knowledge graph (KG), with the stacked memory (SM), and without the memory (w/o Memory). We report mean success rate and their 1 standard errors (SE).

| Mission | # rooms | Metrics | KG | Gemini-2.5-Flash w/o Memory | SM | KG | Gemini-2.5-Pro w/o Memory | SM |
|---|---|---|---|---|---|---|---|---|
| Put Next To | 2×2 | Success (%) | **96.7** ± 3.3 | 90.0 ± 5.6 | 87.7 ± 6.3 | 83.3 ± 6.6 | 73.3 ± 8.2 | **90.0** ± 5.6 |
| | | GED | 4.97 ± 1.19 | — | — | 3.43 ± 1.06 | — | — |
| | 3×3 | Success (%) | **66.7** ± 8.8 | 36.7 ± 8.9 | 36.7 ± 8.9 | **70.0** ± 8.5 | 56.7 ± 9.2 | 43.3 ± 9.2 |
| | | GED | 8.43 ± 1.62 | — | — | 6.33 ± 1.45 | — | — |
| Open Door | 2×2 | Success (%) | **100.0** ± 0.0 | 86.7 ± 6.3 | **100.0** ± 0.0 | **100.0** ± 0.0 | 93.3 ± 4.6 | **100.0** ± 0.0 |
| | | GED | 1.13 ± 0.36 | — | — | 1.60 ± 0.34 | — | — |
| | 3×3 | Success (%) | **70.0** ± 8.5 | 66.7 ± 8.8 | **70.0** ± 8.5 | **83.3** ± 6.8 | 73.3 ± 8.2 | 76.7 ± 7.9 |
| | | GED | 9.70 ± 1.87 | — | — | 3.60 ± 0.82 | — | — |

Table 3: Success rate of `PutNextTo` mission. We compare the heuristic actor and LLM-as-agents to test extendability of our method. While LLM-as-Agent struggles due to inevitable hallucinations, our method was able to solve some tasks.

| # rooms | Metrics | Gemini-2.5-Flash | | Gemini-2.5-Pro | |
|---|---|---|---|---|---|
| | | Heuristic Actor | LLM-as-agent | Heuristic Actor | LLM-as-agent |
| 2×2 | Success (%) | 96.7 ± 3.3 | 20.0 ± 7.4 | 83.3 ± 6.6 | 13.3 ± 6.3 |
| | GED | 4.50 ± 1.65 | 5.33 ± 1.10 | 3.17 ± 1.56 | 8.90 ± 1.13 |
| 3×3 | Success (%) | 66.7 ± 8.8 | 20.0 ± 7.4 | 70.0 ± 8.5 | 10.0 ± 5.6 |
| | GED | 7.80 ± 1.63 | 6.80 ± 1.06 | 6.00 ± 1.83 | 9.2 ± 0.95 |

per room and three entities per room. In total, our experiments cover 20 unique layouts. To assess different memory configurations for the LLM planner, we designed experiments under two operational modes: a dynamic memory setting and a static memory setting. The dynamic setting, which simulates real-world deployment, requires the planner to explicitly call external tools to retrieve information from memory. In contrast, the static setting provides continuous access to the full graph information. Results for the static setup are reported in Appendix E.

To evaluate the efficacy of our knowledge graph as a memory module, we adopted two additional baselines. The first, **without Memory (w/o Memory)**, removes the knowledge graph construction stage entirely. In this setting, the agent still leverages the same DSL for planning and exploration but operates without external memory, serving as a baseline to measure the direct contribution of our framework (**KG**). The second, **Stacked Memory (SM)**, replaces the graph-based memory with a linear, stack-structured alternative to assess the role of memory topology. In this baseline, memory is built sequentially: at each step, the output of the Extract Information module is appended to a linear data store. To accommodate this structure, we implemented three retrieval tools that provide the LLM with action sequences and trajectories to summarize (i) the most recent decision, (ii) historical information about a queried entity, and (iii) historical information about the most recently observed closed or locked door.

## 4.1 EFFECTIVENESS OF EXTERNAL MEMORY UNDER PARTIALLY OBSERVABILITY

We analyze the effectiveness of LLMs in leveraging external modules for memory storage and retrieval in a partially observable environment using a planning DSL. The actor interacting with the environment is instantiated in two variants: (1) an expert heuristic actor, which isolates the contributions of memory and planning from the variability in action execution, and (2) an LLM-as agent, which introduces additional considerations due to the inherent uncertainty of LLM outputs. We evaluate our framework on two BabyAI missions, `PutNextTo` and `OpenDoor`, with results presented in Table 2. To assess the impact of the knowledge graph (KG), we compare against an ablated version of our model without the knowledge graph (w/o Memory). The full framework consistently outperforms this baseline, highlighting the critical role of structured external memory in enabling effective agent behavior.

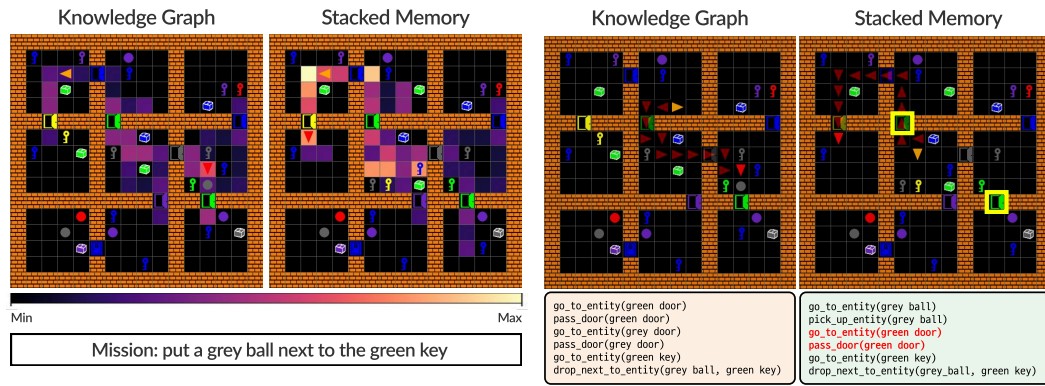

(a) State visitations under Gemini-2.5 Flash planner.    (b) Plans generated by Gemini-2.5 Flash planner.

Figure 5: Comparison of a **knowledge graph (KG)** versus **stacked memory (SM)** for a Gemini-2.5-Flash planner. The KG's structured representation enables efficient exploration **(a)** and results in correct plan **(b)**. In contrast, the SM leads to confused exploration and planning failure, as the agent cannot distinguish between to identical doors (planning errors: red in text, yellow in image).

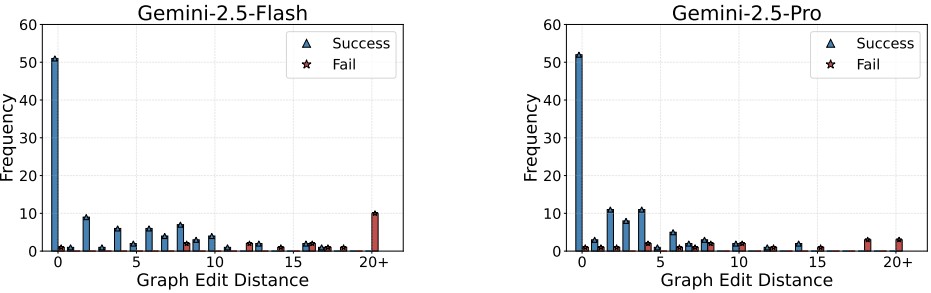

Figure 6: Distribution of Graph Edit Distance (GED) based on Mission Success. The distribution of Knowledge Graph GED scores for task successes (blue triangles) and failures (red stars), using Gemini-2.5-Flash (left) and Gemini-2.5-Pro (right). Overall, successful episodes tend to be concentrated at lower GED values, indicating higher graph accuracy, while failures are more distributed across higher GED values. GED scores exceeding 20 are aggregated into the 20+.

To evaluate the extendability of our framework to non-expert implementations of DSLs, we adopt an LLM-as-agent approach (Paglieri et al., 2024). The results for the `PutNextTo` mission are summarized in Section 4.1. In comparison to the rule-based heuristic actor, LLM-based implementation exhibits a noticeable drop in performance. This outcome is expected, as large language models are prone to hallucination Kalai et al. (2025) and struggles to construct a coherent inner model from egocentric observations Yang et al. (2025). We hypothesize that designing more fine-grained DSLs could help mitigate this limitation, and we view the automatic discovery of such functions as an important direction for future research.

> The success rate of the Gemini-2.5-Flash using only DSL was 63.3%, whereas the rate increased to **81.7%** with the addition of the knowledge graph. This demonstrates that the knowledge graph pipeline enables more effective problem-solving. The detailed execution result can be found in **Appendix D**.

## 4.2 EFFECTIVENESS OF GRAPH-BASED MEMORY STRUCTURE

In this section, we evaluate the effectiveness of a graph-based memory structure by comparing it against a stacked memory alternative to examine whether the structure of external memory influences performance. The agent equipped with stacked memory fails to navigate the environment efficiently. As illustrated in Figure 5a, which visualizes cell visitation frequency, the agent frequently

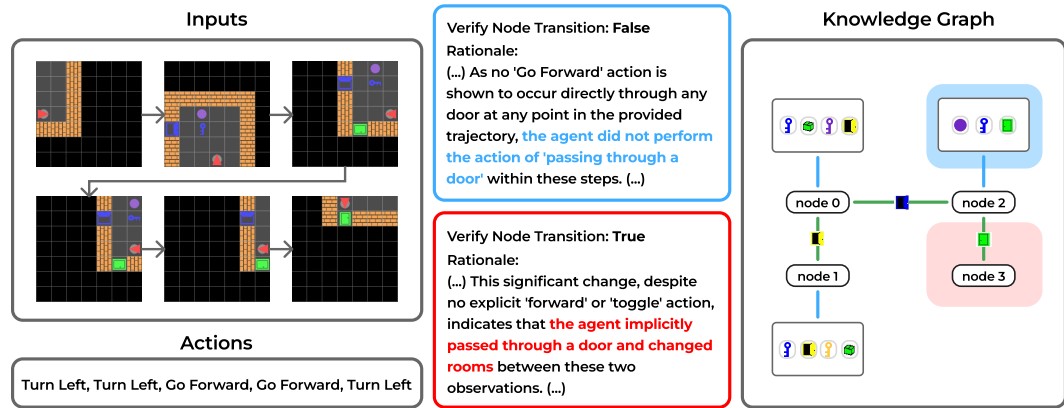

Figure 7: Examples of planner correctly (**Blue Box**) and incorrectly (**Red Box**) verifies whether the agent has transitioned into another node. LLM planner confuses observation changes due to rotation to node changes, leading to spurious node expansion (**node 3**).

revisits already explored cells and repeatedly rediscovers objects. This high rate of revisitation indicates that the stacked memory is not effecively utilized, thereby hindering efficient navigation. In contrast, the knowledge graph-based agent exhibits more structured and efficient exploration patterns, despite the inevitable redundancies caused by partial observability. Moreover, leveraging the knowledge graph-based memory facilitates efficient pathfinding between mission-critical objects, such as "a grey ball" and "a green key".

The inefficiency is further underscored by the suboptimal plans generated by the planner. Figure 5b shows examples of LLM-generated plans along with the corresponding execution trajectories for our method and the baseline. The stacked-memory agent confused the green door in the lower-right, connecting cells 9 and 6, with the green door connecting cells 5 and 2. This confusion suggests that stacked memory is an ineffective strategy for managing a dynamically growing memory. In contrast, the agent equipped with the knowledge graph successfully planned a trajectory to reach the target object (a gray ball).

> Simply using external memory does not guarantee improved performance; in fact, a naive approach can be **detrimental**. Our result show that a graph-based memory improved an agent's efficiency, while a poorly structured memory harmed the performance.

## 4.3 EVALUATING LLM-CONSTRUCTED KNOWLEDGE GRAPHS

A key challenge in Knowledge Graph Construction for sequential decision-making is the consistent identification and tracking of objects over time. Identity errors introduce redundant nodes, reducing both computational and memory efficiency, particularly in dynamic, partially observable environments where an agent actions (e.g., relocating objects induce variability.

To evaluate whether LLM-constructed knowledge graphs provide meaningful support for planning, we adopt graph edit distance (GED) Sanfeliu & Fu (2012) as a proxy for structural similarity to the ground-truth graph. Intuitively, a graph that more closely matches the true environment should enable more accurate reasoning and planning. GED offers a principled way to quantify this similarity through the minimum number of node or edge edit operations required for alignment.

As in Table 2, average GED increases with environment size, rising from $2 \times 2$ to $3 \times 3$ grids due to compounded inference errors. More importantly, Figure 6 demonstrates a strong inverse relationship between GED and task success. In the `PutNextTo` mission, successful trials averaged GED values of 3.62 ($2 \times 2$) and 3.25 ($3 \times 3$), while failed trials averaged 30.0 and 16.9, respectively. When GED was 0, success rates reached 98%, whereas no successful trials occurred with GED $> 20$.

These findings support the intuition that structural fidelity is critical: the closer the constructed graph is to the ground truth, the more useful it becomes for guiding sequential decision-making. Finally,

Figure 7 illustrates typical failure modes. Large observation changes often caused the LLM to misclassify a revisited room as new, leading to spurious graph expansion and degraded accuracy.

> The results demonstrate that the accuracy of the knowledge graph impacts mission success. The success rate ranged from **98.10%** with accurate knowledge graphs to 0% when accumulated errors caused the GED to exceed 20. This suggests that LLMs possess an intrinsic ability to utilize knowledge, and the graph accuracy impact to task performance.

## 5 RELATED WORK

**LLMs as Agents in Sequential Decision-Making**  Large language models (LLMs) have demonstrated strong performance across several challenging tasks, including question answering (Rajpurkar et al., 2016), mathematics (Hendrycks et al., 2021), and, more recently, complex iterative interactions within real-world environments. For instance, Ma et al. (2024) achieved notable results in the real-time strategic decision-making environment StarCraft II by introducing the Chain of Summarization (CoS) method to enhance LLMs' decision-making efficiency. Furthermore, Paglieri et al. (2024) benchmarked LLM-as-agent approaches across several game-based environments, including BabyAI (Carta et al., 2023), TextWorld (Côté et al., 2018), Baba Is AI (Cloos et al., 2024), MiniHack (Samvelyan et al., 2021), and NetHack Learning Environment (NLE) (Küttler et al., 2020). However, LLM-as-agent approaches exhibit limitations in long-context scenarios, particularly in tasks that require effective utilization of historical information.

**Addressing Hallucination via Knowledge Retrieval**  Although the ability of LLMs to handle long contexts has improved, they still suffer from hallucinations—a critical issue in long-context problems, such as sequential interactions with an environment. To mitigate this, prior work has proposed Retrieval-Augmented Generation (RAG) and its variants (Lewis et al., 2020; Yu et al., 2022; Zheng et al., 2023). Han et al. (2024) introduced GraphRAG, which enhances RAG by incorporating graph-based structures. Unlike conventional RAG, GraphRAG operates on graph-structured data characterized by diverse formats and heterogeneous sources. However, these approaches remain constrained by their reliance on retrieving information from static documentation and by their passive dependence on such information. Their primary role is to improve factual grounding by retrieving facts during inference, but they remain passive with respect to environments where the knowledge base itself is incomplete or evolving.

**Retrieving Information from External Memory**  Retrieval-Augmented Generation (RAG) (Lewis et al., 2020) augments LLMs with the ability to retrieve semantically relevant document chunks from an external knowledge base, thereby mitigating hallucinations which is a critical limitation of LLMs when faced with queries that extend beyond their training data or demand up-to-date information. GraphRAG (Han et al., 2024) extends the RAG framework by incorporating graph-structured knowledge representations, enabling more effective retrieval through the exploitation of relational and structural information. In this respect, the concept of constructing a knowledge graph for retrieval aligns closely with our approach.

## 6 CONCLUSION

This paper investigates the application of large language models (LLMs) to navigation tasks in partially observable environments, focusing on how to equip LLMs with mechanisms for memory, reasoning, and planning under uncertainty. We propose a framework that combines domain-specific languages (DSLs) for high-level planning with a dynamically constructed knowledge graph to serve as an external memory. Our approach enables the agent to iteratively plan, act, and update its knowledge, effectively bridging the gap between abstract reasoning and low-level action execution. Experimental results in complex MiniGrid environments demonstrate that leveraging a knowledge graph significantly improves planning efficiency, task success rates, and robustness under partial observability. These findings highlight the potential of combining LLM reasoning with structured, adaptive memory representations, suggesting a promising direction for future research in long-horizon, memory-intensive tasks in real-world settings.

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

## A MODULE DETAIL WITH PROMPT

### A.1 KNOWLEDGE GRAPH-BASED MEMORY

**Verify Node Transition Module** In the *Verify Node Transition*, we prompt a Large Language Model (LLM) with the state-action trajectory and a guiding instruction for reasoning. The LLM analyzes this information to determine the validity of the transition. Recognizing the critical impact of this judgment on the knowledge graph's accuracy, we exclusively employ an ensemble method for this module. The node transition is determined by a majority vote over five trials. To verify a node transition, we query an LLM to check the transition's occurrence and predict the necessary connection information for the subsequent node. To enhance the reliability of the response, we incorporate a self-evaluation mechanism inspired by the Independent Evaluation method Yao et al. (2023).

```
Verify Node Transition Prompt

[TEXT INPUT]

{Room_Description}
{room_description}
{/Room_Description}

{Action_Description}
{action_description}
{/Action_Description}

<Trajectories>
<Step_num>
{Trajectory}
{Direction}
</Step_num>

<Action_num>
{action}
</Action_num>

...

</Trajectories>

<Connected_Door_Instruction>
{connected_door_instruction}
</Connected_Door_Instruction>

<Door_Direction_Instruction>
{door_direction_instruction}
</Door_Direction_Instruction>

<Check_Pass_Door_Instruction>
{verify_node_transition_instruction}
</Check_Pass_Door_Instruction>

[IMAGE INPUT]

{images}

----------------------------------------

[OUTPUT]

'''LLM Reasoning'''
{
    connected_door_rationale: {rationale}
    connected_door: {connected_door}
    door_direction_rationale: {rationale}
    door_direction: {door_direction}
    check_pass_door_rationale: {rationale}
    check_pass_door: {check_pass_door}
    answer_confidence_score: {answer_confidence_score}
}
```

**Extract Information**    In the *Extract Information*, the LLM is prompted to summarize the current observation. It is provided with the observation and guided by instructions for reasoning, analyzing the observation to generate a summary containing only the most critical information. This prompting strategy serves the two purposes of enabling effective differentiation between graph nodes and ensuring efficient memory utilization by storing only essential information. Subsequently, the LLM's responses contain both underscores and spaces, all underscores are converted to spaces for consistent formatting.

---

**Extract Information Prompt**

```
[TEXT INPUT]

<Observation>
{observation}
</Observation>

<Entity_Listing_Instructions>
{entity_listing_instructions}
</Entity_Listing_Instructions>

<Current_Room_Entities_Instructions>
{current_room_entities_instructions}
</Current_Room_Entities_Instructions>

<Current_Room_Entities_Relationships_Instructions>
{current_room_entities_relationships_instructions}
</Current_Room_Entities_Relationships_Instructions>

<Direction_Of_Entities_Instructions>
{direction_of_entities_instructions}
</Direction_Of_Entities_Instructions>

[IMAGE INPUT]

{image}

---------------------------------------

[OUTPUT]

'''LLM Reasoning'''
{
    current_room_entities_rationale: {rationale}
    current_room_entities: {current_room_entities}
    entities_relationships_rationale: {rationale}
    entities_relationships: {entities_relationships}
    direction_of_entities_rationale: {rationale}
    direction_of_entities: {direction_of_entities}
}
```

**Determine the Current Node** In the *Determine the Current Node*, the LLM determines the current node. It is provided with information about the previously occupied node, the nodes connected to that previous node, and the current observation. The LLM determines whether the current node is a previously visited node or unvisited node, and it responds with the corresponding graph node number. The selection of a graph-based localization method over a coordinate-based approach was driven by the potential for compounding errors when requiring an LLM to manage memory. This memory is intended to mitigate significant error accumulation. The effectiveness of this approach is supported by the GED experiment results. Furthermore, the knowledge graph facilitates the efficient storage of entity information.

---

**Determine the Current Node Prompt**

```
[TEXT INPUT]

<Observation>
{observation}
</Observation>

<Observed_Entities>
{observed_entities}
</Observed_Entities>

<Room_Information>
- Previous_Room_Number: {previous_room_number}
<Connected_with_Previous_Rooms>
- The previous room is connected with {connection_information}.
- Room contains the entities: {entities_information}
</Connected_with_Previous_Rooms>
</Room_Information>

<Rooms_List>
- {nodes}
</Rooms_List>

<Description>
{description}
</Description>

<Current_Graph_Node_Instructions>
{current_graph_node_instructions}
</Current_Graph_Node_Instructions>

----------------------------------------

[OUTPUT]

'''LLM Reasoning'''
{
    current_graph_node_id_rationale: {rationale}
    current_graph_node_id: {current_graph_node_id}
}
```

**Update Entity Information**  In the *Update Entity Information*, the LLM provides both the graph information and the current observation to synthesize previously observed entity information with the current observation. the environment contains visually identical entities, external information is required to differentiate entities. To enable the LLM to distinguish between these entities, we provided the relational and directional information. The LLM responds with the aggregated observation, including updated relations and directions. Subsequently, we applied a post-processing to convert all underscores in the entity information to spaces.

---

**Update Entity Information Prompt**

```
[TEXT INPUT]

<Current_Room_Entities>
<Entity_List>
{node_entitiy_list}
</Entity_List>
<Entities_Relationships>
{node_entities_relationships_information}
</Entities_Relationships>
<Direction_Of_Entities>
{node_direction_of_entities_information}
</Direction_Of_Entities>
</Current_Room_Entities>

<Currently_Partial_Observed_Entity_Information>
<Entity_List>
{entitiy_list}
</Entity_List>
<Entities_Relationships>
{entities_relationships_information}
</Entities_Relationships>
<Direction_Of_Entities>
{direction_of_entities_information}
</Direction_Of_Entities>
</Currently_Partial_Observed_Entity_Information>

<Door_Change>
{door_change_information}
</Door_Change>

<Inventory_Change>
{inventory_change_information}
</Inventory_Change>

<Instruction>
{instruction}
</Instruction>

----------------------------------------

[OUTPUT]

'''LLM Reasoning'''
{
    graph_nodes_entities_rationale: {rationale}
    graph_nodes_entities: {entities}
    graph_nodes_entities_relationships_rationale: {rationale}
    graph_nodes_entities_relationships: {entities_relationships}
    graph_nodes_direction_of_entities_rationale: {rationale}
    graph_nodes_direction_of_entities: {direction_of_entities}
}
```

## A.2 PLANNING USING DOMAIN-SPECIFIC LANGUAGE

**Plan with DSLs**  In the *Plan with DSLs*, the LLM receives as input the current inventory, the previous plan and its execution status, the current observation, the agent's facing direction, and the set of available DSL instructions with their descriptions. Conditioned on this information, the LLM generates a sequence of DSL instructions, accompanied by a rationale, that aligns with its high-level plan for solving the mission. The generation process leverages the knowledge graph through predefined tool calls invoked by the LLM's decisions. In addition, the LLM specifies a target entity for the plan, together with a rationale, indicating the object on which the current plan should focus.

---

### Plan with DSLs Prompt

```
[TEXT INPUT]

<Rule_Description>
{rule_description}
</Rule_Description>

<Graph_Information>
{graph_information}
</Graph_Information>

<Subplan_Target_Entity_Instructions>
{subplan_target_entity_instructions}
</Subplan_Target_Entity_Instructions>

<Subplans_Instructions>
{subplans_instructions}
</Subplans_Instructions>

<Inventory>
{inventory}
</Inventory>

<Last_Plan>
{last_plan}
</Last_Plan>

<Last_Plan_Completion>
{last_plan_completion}
</Last_Plan_Completion>

<Facing_Direction>
{facing_direction}
</Facing_Direction>

<DSL_List>
{dsl_list}
</DSL_List>

[IMAGE INPUT]

{image}

----------------------------------------

[OUTPUT]

'''LLM Reasoning'''
{
    subplan_target_entity_rationale: {rationale}
    subplan_target_entity: {subplan_target_entity}
    subplans_rationale: {rationale}
    subplans: {subplans}
}
```

**Verify Completion**   In the *Verify Completion*, the LLM determines whether the previous plan has been completed and provides a rationale for its judgment. This decision is based on the agent's current information, including its inventory, door traversal status, facing direction, current observation, the previous plan, and the number of times that plan has been repeated. In addition, the LLM evaluates whether the plan should be adjusted—and explains why—if it has remained incomplete for an extended period.

```
Verify Completion Prompt

[TEXT INPUT]

<Graph_Information>
{graph_information}
</Graph_Information>

<Inventory> {inventory} </Inventory>

<Pass_Door>
{pass_door_information}
</Pass_Door>

<Observation>
{observation}
</Observation>

<Facing_Direction> {facing_direction} </Facing_Direction>

<Check_DSL_Commands>
{check_DSL_commands}
</Check_DSL_Commands>

<Previous_Plans> {previous_plans} </Previous_Plans>

<Last_Plan> {last_plan} </Last_Plan>

<Num_Repeats_Last_Plan>
{num_repeats_last_plan}
</Num_Repeats_Last_Plan>

<Is_Complete_Instruction>
{is_complete_instruction}
</Is_Complete_Instruction>

<Need_To_Adjust_Instruction>
{need_to_adjust}
</Need_To_Adjust_Instruction>

[IMAGE INPUT]

{image}

----------------------------------------

[OUTPUT]

'''LLM Reasoning'''
{
    is_complete_rationale: {rationale}
    is_complete: {is_complete}
    need_to_adjust_rationale: {rationale}
    need_to_adjust: {need_to_adjust}
}
```

**Execute** In the *Execute*, the LLM-as-agent, acting as the actor, analyzes the mission, the current observation, and its facing direction, and generates up to 10 low-level actions in a single turn, accompanied by a rationale. The instruction prompt supplies the LLM-as-agent with the available action set, the transition dynamics of the environment, and a concise guideline on how to handle blockers when encountered.

---

**Execute Prompt**

```
[TEXT INPUT]

<Rule_Description>
{rule_description}
</Rule_Description>

<Action_Description>
{action_description}
</Action_Description>

<Mission_Description>
{mission_description}
</Mission_Description>

<Graph_Information>
{graph_information}
</Graph_Information>

<Mission>
{subplan}
</Mission>

<Observation>
{problem}
</Observation>

<Direction>
You are facing north.
</Direction>

<Inventory>
{inventory}
</Inventory>

<Instructions>
{instructions}
</Instructions>

[IMAGE INPUT]

{image}

----------------------------------------

[OUTPUT]

'''LLM Reasoning'''
{
    actions_rationale: {rationale}
    actions: {actions}
}
```

---

## A.3 Tool Call

**Get Neighbor Entity Information**    Designed for short-term planning, it operates by receiving a node number as a parameter to return a string with all information about the specified node and its neighbors. Since the LLM cannot natively determine if all nodes have been visited, the tool also provides a visitation count for each node within the current decision step to inform the agent's exploration strategy.

**Search Closest Entity**    It receives a node and a target entity as parameters, performs a Breadth-First Search (BFS), and provides information on the nearest node containing that entity. If the entity is not present in memory, it returns a information that the entity has not been discovered. Conversely, if the nearest entity is found, it provides the sequence of node transitions required to reach it. This information helps the agent determine whether it needs to perform further exploration or formulate a long-term plan.

**Find Unexplored Closed Door**    It is designed to find the shortest path to the nearest closed or locked door from a given node. It receives the node as a parameter and performs a Breadth-First Search (BFS). If no such door is found in the memory, it returns a notification that the closed or locked door is undiscovered. Otherwise, it returns the sequence of node transitions that constitutes the path to the nearest closed or locked door. This information enables the agent to formulate long term exploration plans.

## B  ENVIRONMENT DETAILS

We extend BabyAI (Chevalier-Boisvert et al., 2019), a partially observable 2D gridworld simulation. Built on the MiniGrid platform, BabyAI supports efficient simulation and offers a range of instruction-following tasks using a simplified synthetic language called Baby Language. Each layout consists of $n$ rooms connected by colored doors, with objects placed throughout. Objects are defined by color and type. While unlocked doors can be opened freely, locked doors require keys of the matching color. At each time step, the agent receives a partial observation representing its $7 \times 7$ field of view. Walls and doors obstruct the observation, even when doors are open.

The environment provides observations in two modalities: pixel-based images and textual descriptions. While BabyAI offers default image-rendered assets, we modify the object assets in the pixel-based observations to enhance visual distinctiveness and improve object recognition by LLMs. The textual representation encodes each cell using predefined object descriptors (e.g., Wall, Yellow_Closed_Door, Blue_Box), separated by semicolons, enabling precise symbolic reasoning over the observed grid. This structured format enables symbolic reasoning over spatial configurations while preserving compatibility with language-based models. The action space supports six actions: `Go Forward`, `Turn Left`, `Turn Right`, `Pickup`, `Drop`, and `Toggle`. The `Toggle` action allows the agent to interact with doors, such as opening, closing, or unlocking them.

## C  ABLATION STUDY ON OBSERVATION MODALITIES

Table 4 shows the experimental results comparing performance when the agent receives environmental information as text-only versus when image observations are also provided. When using only the text observation, the agent succeeded in 38 out of 60 trials. In contrast, when image observations were added, the agent succeeded in 49 out of 60 trials. This suggests the LLM achieves a better understanding of the environment, as it can leverage the additional information from the images.

Table 4: Performance comparison between text-only and text with image on PutNextTo missions. We evaluate the performance of the knowledge graph approach using two different observation.

| Observation Format | # rooms | Accuracy (%) |
|---|---|---|
| Image & Text | 2×2 | 96.7 ± 3.3 |
|  | 3×3 | 66.7 ± 8.6 |
| Text | 2×2 | 83.3 ± 6.6 |
|  | 3×3 | 43.3 ± 9.0 |

# D    EXAMPLE CASE

We visualize the subplans produced by the LLM planner augmented with a LLM-generated knowledge graph, alongo with the full execution trajectory of a downstream actor, in Figure 8 and Figure 9.

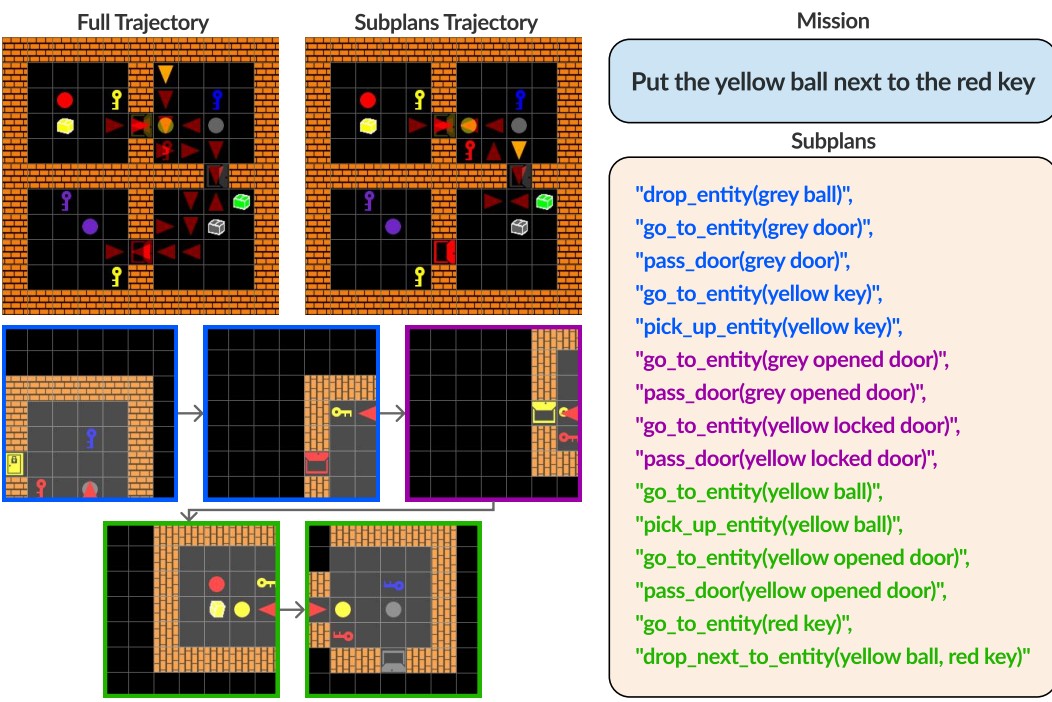

Figure 8: The LLM planner augmented with a knowledge graph successfully generates a long, coherent sequence of subplans to accomplish the mission: "Put the yellow ball next to the red key." The agent moves from the start position (▶) to the final position (▶). The LLM generated entire sequence of subplans at once, demonstrating its capabilities for long-horizon reasoning in partially observable environments.

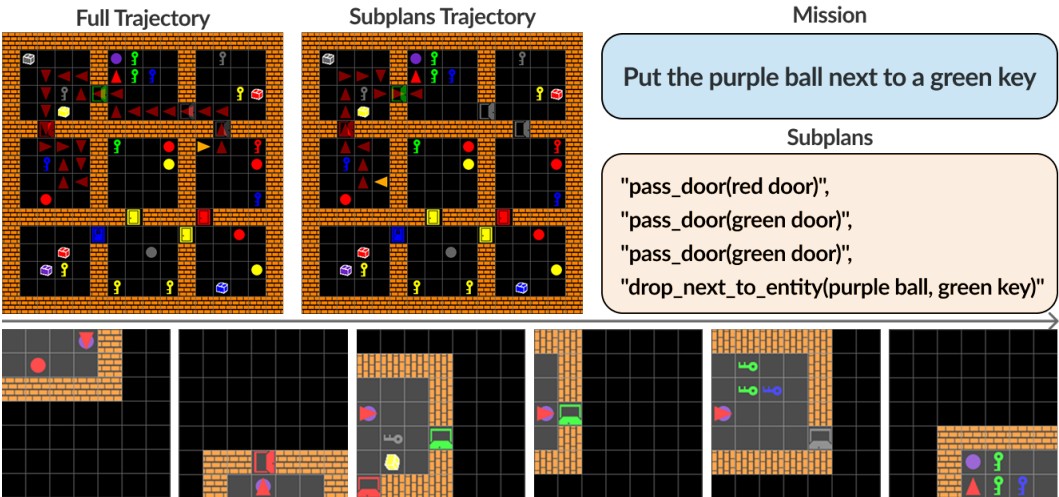

Figure 9: The LLM planner, using a knowledge graph, creates a long and logical series of subplans to complete the mission: "Put the purple ball next to a green key." The agent moves from the start position (▶) to the final position (▶).

We further highlight the failure modes of LLM planner augmented with stacked memory, where every attempts fails to complete the mission, as shown in Figure 10 and Figure 11.

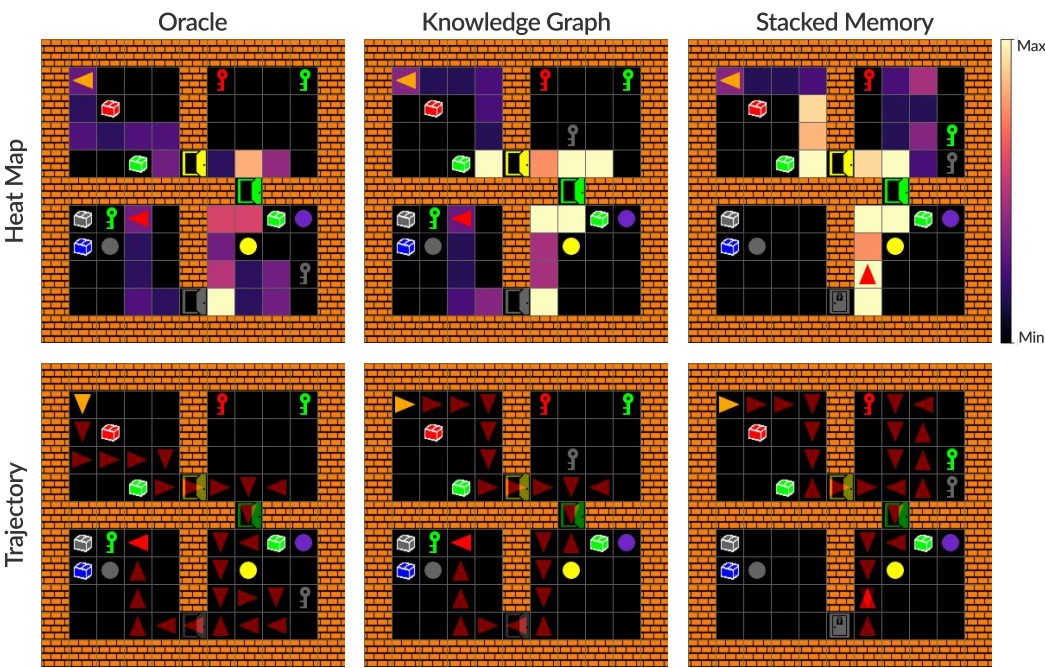

Figure 10: The visitation heat map and the trajectory of oracle agent, knowledge graph-augmented agent, and stacked memory-augmented agent.

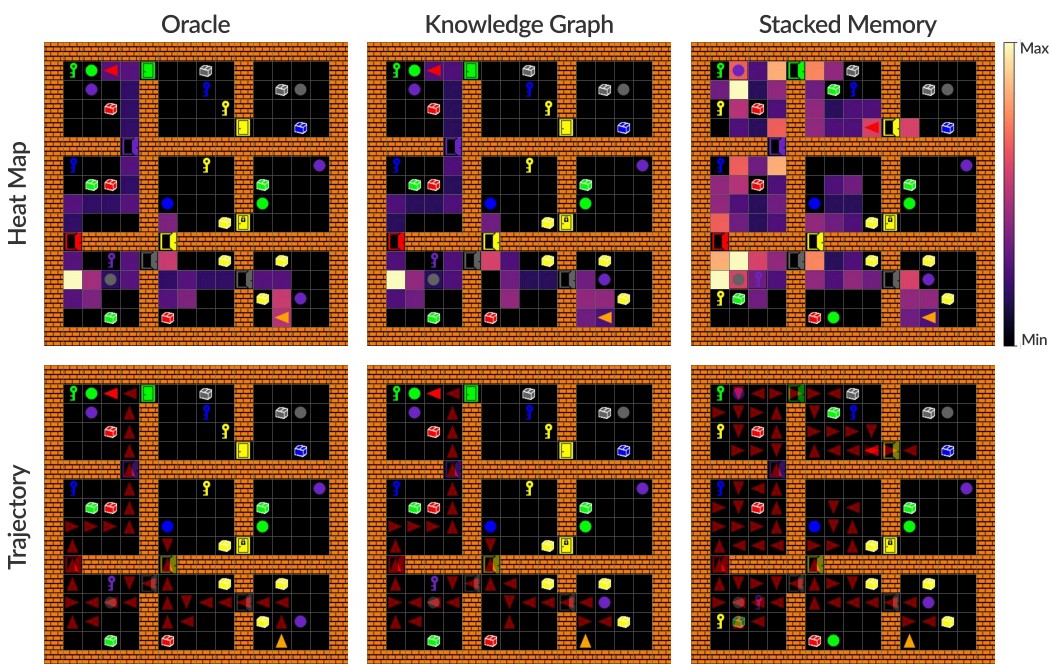

Figure 11: The visitation heat map and the trajectory of oracle agent, knowledge graph-augmented agent, and stacked memory-augmented agent.

# E    COMPARISON OF MEMORY ACCESS METHODS

To evaluate the LLM's ability to handle the dynamics of information gathering and utilizing in sequential decision-making, we compare two settings: **dynamic memory** where the model performs tool-calling experiments, and **static memory**, where the entire knowledge graph is provided in the context window.

The results are summarized in Table 5. Although static memory provides the LLM with more information at each step, its performance was lower. This finding is consistent with the comparison between knowledge graph and stacked memory, suggesting that inefficient memory structures can hinder the performance.

Table 5: Success rate of `PutNextTo` mission. We compare dynamic memory, where the agent controls the tool calls, and static memory, where all information are always given.

| # rooms | Metrics | Gemini-2.5-Flash | |
| --- | --- | --- | --- |
| | | Dynamic Memory | Static Memory |
| 2×2 | Success (%) | $96.7 \pm 3.3$ | $93.3 \pm 4.8$ |
| | GED | $4.50 \pm 1.65$ | $2.79 \pm 1.09$ |
| 3×3 | Success (%) | $66.7 \pm 8.6$ | $58.6 \pm 9.03$ |
| | GED | $7.80 \pm 1.63$ | $9.38 \pm 2.17$ |

