# OpenReview forum: "GraphMind: LLMs as Dynamic Knowledge Builders for Sequential Decision-Making"
_ICLR.cc/2026/Conference — ICLR 2026 Conference Withdrawn Submission_

### Official Review · Reviewer_us7X · 2025-10-31

**Soundness:** 2
**Presentation:** 3
**Contribution:** 1
**Rating:** 2
**Confidence:** 3

**Summary:**

"GraphMind: LLMs as Dynamic Knowledge Builders for Sequential Decision-Making" proposes an LLM agent that constructs and uses a dynamic knowledge graph (KG) as an external memory for planning in partially observable BabyAI environments. The agent alternates between graph construction (building nodes/edges from observations) and structured planning via a domain-specific language (DSL). Experiments on small BabyAI gridworlds (maps with 2×2 and 3×3 rooms) with tasks like OpenDoor and PutNextTo suggest that the graph-based memory improves task success rates over a “stacked memory” baseline and a no-memory setup.

**Strengths:**

* The framework is well-engineered, with detailed modular prompts and clear visualization of how the knowledge graph evolves.
* The combination of structured memory and DSL-based planning is conceptually clean and helps connect symbolic reasoning with embodied action.
 * The experiments, though small-scale, are thorough within the limited setting, including GED-based analysis of graph accuracy and qualitative visualizations.

**Weaknesses:**

* **Novelty**: The idea of dynamically constructing and exploiting a knowledge graph as an LLM memory in partially observable settings is not new. Prior work like AriGraph explored very similar designs: LLMs building and querying graph-structured memory for decision-making in POMDP. The paper’s core claim of being the “first” dynamic KG-based planner is overstated.
 * **Evaluation scope**: The empirical validation is extremely limited—only two BabyAI missions (OpenDoor, PutNextTo) in 2×2 and 3×3 grids, with about 20 layouts and three trials each. There is no testing on larger, more complex environments, distractors, or longer object-interaction chains. Claims of generality to “real-world” or “long-horizon” reasoning are unsubstantiated.
* **Baselines**: Comparisons are weak—only a simple “stacked memory” and a no-memory version. Missing comparisons with prior KG/RAG-based or agentic-memory frameworks makes it unclear how much progress is achieved.
* **Practicality and efficiency**: The paper omits any discussion of runtime, compute, or token costs, which are crucial for assessing the feasibility of maintaining dynamic KGs with LLMs.

**Questions:**

Please address the weaknesses mentioned in the previous section.

1. Could you clarify how your approach fundamentally differs from prior work such as AriGraph or other KG-based memory frameworks for LLM-driven agents?

2. Could you elaborate on why experiments are limited to small BabyAI environments (2×2, 3×3) and discuss how you expect the method to scale to larger or more complex tasks?

3. Please provide quantitative data on computational and token-level costs (e.g., inference time per step, prompt length, graph update overhead) to assess the scalability and practicality of maintaining dynamic KGs in your setup

**Details Of Ethics Concerns:**

No ethics concerns with respect to this work.

---

> ### Author Response · Authors · 2025-11-25
>
> We thank the reviewer for the thorough and constructive comments. We hope we can address your concerns below.
>
> **Q1. Could you clarify how your approach fundamentally differs from prior work such as AriGraph or other KG-based memory frameworks for LLM-driven agents?**
>
> We appreciate your suggestion to compare with similar prior work. Our goal is to solve navigation tasks in partially observable environments that provide only minimal information, such as objects and the main goal, by constructing a knowledge-graph memory using the LLM’s reasoning ability with minimal external assistance. While AriGraph [1] addresses partial observability by building semantic and episodic knowledge graphs, it assumes that the environment provides more information, such as global locations and the valid action set.
>
> While AriGraph constructs triplets using the current location and identifies unexplored areas with a hand-coded algorithm, our LLM agent must track and infer the actor’s current location and maintain spatial map information. Furthermore, we provide a framework in which the LLM agent plans actions suited to the current state across the entire DSL, whereas AriGraph selects only from a valid action set provided by the environment. This demonstrates that our method can be effectively extended to more general applications.
>
> **Q2. Could you elaborate on why experiments are limited to small BabyAI environments (2×2, 3×3) and discuss how you expect the method to scale to larger or more complex tasks?**
>
> We focus on enhancing LLMs’ planning capabilities in partially observable environments that provide only limited local information. BabyAI is well-suited for this purpose, as it is a partially observable 2D gridworld combining object manipulation, navigation, and mission execution. We designed four-room and nine-room layouts, each consisting of 25 cells (excluding walls), and further increased partial observability by blocking sight even when doors are open. The nine-room (3×3) layout corresponds to the BossLevel, the most challenging level in BabyAI, where, as shown in Table 2, LLMs struggle to solve the tasks. Therefore, we believe that evaluation in this setting is sufficient to assess LLMs’ planning capabilities.
>
> It is true that the success rate decreases in larger, more complex environments that require more steps, due to cumulative errors. However, GraphMind, where the LLM agent corrects its own errors using modules such as Verify Transition and Revise Plan, demonstrated strong performance in the nine-room (3×3) layout.
>
> **Q3. Please provide quantitative data on computational and token-level costs (e.g., inference time per step, prompt length, graph update overhead) to assess the scalability and practicality of maintaining dynamic KGs in your setup.**
>
> We compare the token-level costs of different memory structures across our method (dynamic KG) and the baselines: stacked memory (SM), which uses a linear, stack-structured alternative, and static memory (static KG), where the entire knowledge graph is provided in the context window. We are expecting that our method improves the agent’s performance with minimal cost. We will update the results as soon as they are available.
>
> [1] Anokhin, Petr, et al. "Arigraph: Learning knowledge graph world models with episodic memory for llm agents." arXiv preprint arXiv:2407.04363 (2024).

---

> > ### Author Response · Authors · 2025-11-27
> >
> > We present the token-level costs associated with different memory structures. The experiments were conducted using Gemini-2.5-Flash models as the planner, with heuristic bots acting as the downstream actors. The values reported below represent the number of tokens consumed per decision step, averaged over five independent runs.
> >
> > Please note the following regarding the experimental setup:
> >
> > 1. As Stacked Memory (SM) does not utilize a KG, costs associated with KG-specific prompts are not applicable (denoted as “-”).
> > 2. “Verify Node Transition” module utilizes an ensemble of LLMs. The reported cost assumes a single LLM; total cost scales linearly with the ensemble size.
> >
> > | Task | Ours | Stacked Memory (SM) | Static KG |
> > | --- | --- | --- | --- |
> > | Verify Node Transition | 3761.96 | - | 3949.38 |
> > | Determine the Current Node | 286.32 | - | 359.01 |
> > | Extract Information | 1557.91 | 1579.28 | 1571.01 |
> > | Plan with DSLs | 2354.95 | 2188.42 | 720.98 |
> > | Verify Completion | 1212.96 | 1148.71 | 1248.21 |
> > | Update Entity Information | 1613.03 | - | 1707.11 |
> > | Total Tokens | 10787.13 | 4916.41 | 9555.70 |
> > | Total Costs ($) | 0.003 | 0.001 | 0.003 |
> >
> > Additionally, the model generates the following number of output tokens per decision step:
> >
> > | Task | Ours | Stacked Memory (SM) | Static KG |
> > | --- | --- | --- | --- |
> > | Verify Node Transition | 1687.30 | - | 1745.88 |
> > | Determine the Current Node | 67.78 | - | 79.17 |
> > | Extract Information | 491.34 | 509.95 | 490.93 |
> > | Plan with DSLs | 102.49 | 124.79 | 90.36 |
> > | Verify Completion | 160.85 | 179.35 | 162.79 |
> > | Update Entity Information | 764.07 | - | 845.10 |
> > | Total Tokens | 3273.83 | 814.09 | 3414.23 |
> > | Total Costs ($) | 0.008 | 0.002 | 0.009 |
> >
> > In our experiment, the average number of decision steps required by our framework was 33.33 for $2 \times 2$ layouts and 46.1 for $3 \times 3$ layouts. In contrast, the SM baseline required 46.93 and 70.43 steps, respectively.
> >
> > This demonstrates a key trade-off: while the per-step cost of our method is slightly higher due to the overhead of managing the KG, **the total trajectory cost** remains comparable. This is evident in more complex problems ($3\times3$ layouts), where our method’s ability to solve tasks in fewer steps offsets the higher per-step token consumption.
> >
> > Furthermore, despite the comparable total trajectory cost, our framework outperforms other baselines in a complex environment (Put Next To, $3\times3$):
> >
> > |  | dynamic KG | SM | static KG |
> > | --- | --- | --- | --- |
> > | Success (%) | 66.7 +- 8.8 | 36.7 +- 8.9 | 57.6 +- 9.03 |

---

### Official Review · Reviewer_1GUm · 2025-11-01

**Soundness:** 3
**Presentation:** 4
**Contribution:** 3
**Rating:** 6
**Confidence:** 4

**Summary:**

The authors present an approach to us a knowledge graph as a memory for reasoning with an LLM in tasks that have longer range dependencies.

**Strengths:**

* I like the setup of the task, which is such that the agent must have a long-range memory in order to find a good solution. It remains a bit unclear how hard the actual tasks are because of the generation and filtering approach.
* The overall results show that the method results in an overall better performance.
* The method is rather intuitive

**Weaknesses:**

* I think the opening premise of the abstract is a bit strange: "the reasoning capabilities of large language models (LLMs) have advanced considerably due to their extensive internal knowledge". This is actually not really a proven thing. The reasoning capabilities seem to stem from analogies with knowledge, but it is, despite anecdotal evidence, not formally proven.

* This work might suffer from a confirmation bias. It is not really possible to say that an LLM could in no way solve these problems; it might just be that we have not yet found the right way to prompt it. A theoretical proof of such a limitation would be a much stronger contribution. In this context it is important that the paper adds computational tools, specifically BFS to the LLMs capabilities. It would be great if one can proof that the LLM cannot perform a BFS of such depth with its context window; but I think it could. It would also be useful to get statistics on how often these tools are called, and how deep the BFS needs to go. As a sub-comment, I am not sure why the details on the tool calls are hidden in the appendix, while they are actually pretty essential.

**Questions:**

* Is it possible for your agent to not store information in the knowledge graph and instead use the context window?
* Do you have a way to measure the accuracy of the content in the knowledge graph compared to what really is in the environment?
* In some places, you call your environment dynamic, but I don't understand what is dynamic in your environment, can you elaborate?

---

> ### Author Response · Authors · 2025-11-25
>
> Thank you for your comprehensive and helpful review.
>
> **Q1. The opening premise of the abstract is a bit strange.**
>
> We will update our statements in the revised manuscript.
>
> **Q2. This work might suffer from a confirmation bias.**
>
> We appreciate your constructive suggestion. There are ongoing controversies about whether LLMs can reliably execute algorithms on their own [1, 2]. In particular, without specialized methods [2], LLM performance degrades as problem size increases. Therefore, we provide BFS as a computational tool, as it is guaranteed to work for problems of any size. We will also move the details on tool calls from the appendix into the main text.
>
> **Q3. Is it possible for your agent to not store information in the knowledge graph and instead use the context window?**
>
> In Table 2, stacked memory (SM) struggles with spatial information, such as rotations, and performs poorly due to cumulative errors when provided with large amounts of context. To address this, we construct a knowledge-graph-based memory structure with cyclical iterations to effectively refine and store relevant information in sequential decision-making tasks under partial observability. Our framework demonstrates stronger performance than the linear, stack-structured memory in complex grid layouts and challenging missions.
>
> **Q4. Do you have a way to measure the accuracy of the content in the knowledge graph compared to what really is in the environment?**
>
> We use graph edit distance (GED) to evaluate the knowledge graphs constructed by the LLM agent. GED is computed by comparing the agent’s trajectory to a ground truth generated using the global locations of objects along the agent’s path.
>
> **Q5. In some places, you call your environment dynamic, but I don't understand what is dynamic in your environment, can you elaborate?**
>
> We used the term “dynamic” to indicate that BabyAI can change through interactions, such as picking up or placing objects and opening or closing doors. We will revise our scripts accordingly.
>
> [1] Shojaee, P., et al. "The Illusion of Thinking: Understanding the Strengths and Limitations of Reasoning Models via the Lens of Problem Complexity. Apple." 2025,
>
> [2] Meyerson, Elliot, et al. "Solving a Million-Step LLM Task with Zero Errors." arXiv preprint arXiv:2511.09030 (2025).

---

### Official Review · Reviewer_dc4W · 2025-11-03

**Soundness:** 3
**Presentation:** 3
**Contribution:** 2
**Rating:** 4
**Confidence:** 3

**Summary:**

The paper proposes GraphMind, an LLM-based agent architecture for sequential decision-making in partially observable environments. It integrates a knowledge graph (KG) as dynamic memory, incrementally built from object interactions to support long-horizon planning. The LLM retrieves relevant KG subgraphs to generate high-level actions, refined into low-level steps. Evaluations focus on custom grid-world navigation tasks (e.g., object collection with partial observability), claiming superior success rates and efficiency over baselines like ReAct and naive LLM planners, especially in exploration-heavy scenarios.

**Strengths:**

The work addresses a pertinent challenge in LLM agents: managing long-term memory, where naive prompting fails due to context limits. The KG as structured memory is a reasonable extension, enabling interpretable retrieval (e.g., via subgraph queries) and reducing hallucination risks. Experiments show intuitive visualizations and quantitative gains, with ablations on memory types providing some insight.

**Weaknesses:**

Despite its aims, GraphMind lacks substantial novelty, largely recombining existing ideas: LLM prompting for planning, KG for memory augmentation, and rule-based updates in grid worlds. KG construction is simplistic (e.g., object-centric heuristics), without learned mechanisms or handling of noisy perceptions, limiting generalization beyond toys.

Experiments are severely constrained: custom, low-dimensional grid tasks ignore standard benchmarks (e.g., MiniGrid, BabyAI), and baselines are weak—missing SOTA like Voyager.

Claims of "dynamic knowledge builders" are vague without ablation on LLM sensitivity (e.g., GPT-4 vs. open models) or scaling to larger graphs (potential explosion).

Robustness tests are contrived (e.g., fixed obstacles), overlooking real shifts like dynamics changes or multi-agent interactions. Overall, the method feels incremental and underexplored, with no theoretical analysis of efficiency or failure modes.

While LLM+KG integration is promising for memory in agents, GraphMind offers no groundbreaking advances. The narrow, toy-like evaluations fail to demonstrate broad impact, and overstated claims undermine credibility.

**Questions:**

In the end of Section 3, authors mentioned "This cyclical structure ensures that planning remains adaptive, resilient to execution errors, and
robust under partial observability." Regarding the partial observability, are there any specific techniques leveraged to tackle the challenge, or just rely on LLMs to provide additional information?

---

> ### Author Response · Authors · 2025-11-25
>
> We sincerely appreciate the reviewer’s feedback.
>
> **Q1. GraphMind lacks substantial novelty, largely recombining existing ideas**
>
> We acknowledge that individual components, such as prompting-based planning and knowledge graphs, are established techniques. However, our primary contribution lies in the **novel integration and the specific implementation** of these methods. We demonstrate that orchestrating these components effectively is non-trivial and essential for achieving robust performance on novel complex tasks and environments.
>
> **Q2. Experiments are severely constrained**
>
> Despite BabyAI’s simplicity, it is a capable environment for evaluating the LLM agent’s planning ability. It is known that LLMs still struggle in seemingly simple environments, such as classical Blocks World and its variants [1]. From that perspective, limiting domains to BabyAI allows us to control variables affecting the evaluation more tightly, while not losing the evaluation capability.
>
> Furthermore, while recent LLM agent framework, such as Voyager, shows strong performance in Minecraft, this is specifically targeted at open-ended lifelong learning agents. Therefore, a direct comparison with such baselines is not straightforward, as our work provides a method for effectively aggregating partial information and planning with it, in a limited number of interactions, a single episode.
>
> **Q3. Claims of "dynamic knowledge builders" are vague**
>
> We use the term “dynamic” to indicate that, in our framework, the LLM-as-agent autonomously updates necessary information and expands its memory through its own planning and judgment. We evaluate two closed models (Gemini-2.5-Flash and Gemini-2.5-Pro), which differ in their reasoning capabilities [2], across various grid layout configurations, demonstrating that our method effectively solves navigation tasks under partial observability.
>
> **Q4. Regarding the partial observability, are there any specific techniques leveraged to tackle the challenge, or just rely on LLMs to provide additional information?**
>
> To solve navigation tasks under partial observability, we employ a knowledge graph as the agent’s memory. This allows the agent to track information from past observations and distinguish identical entities using the four relation types we designed (discussed in Section 3.1). The cyclical structure enables continuous memory updating, verification, and replanning through our modules (Section 3.2). Moreover, we allow LLM agents to call tools to efficiently gather necessary information based on their own judgment, rather than providing all information directly in the prompt. This enables the agent to structurally compensate for information missing due to partial observability.
>
> [1] Valmeekam, K., Stechly, K., & Kambhampati, S. (2024). LLMs Still Can’t Plan; Can LRMs? A Preliminary Evaluation of OpenAI’s o1 on PlanBench (No. arXiv:2409.13373). arXiv. https://doi.org/10.48550/arXiv.2409.13373
>
> [2] Gheorghe Comanici, Eric Bieber, Mike Schaekermann, Ice Pasupat, Noveen Sachdeva, Inderjit Dhillon, Marcel Blistein, Ori Ram, Dan Zhang, Evan Rosen, et al. Gemini 2.5: Pushing the frontier with advanced reasoning, multimodality, long context, and next generation agentic capabilities. arXiv preprint arXiv:2507.06261, 2025.

---

### Official Review · Reviewer_yhK8 · 2025-11-11

**Soundness:** 2
**Presentation:** 3
**Contribution:** 2
**Rating:** 2
**Confidence:** 3

**Summary:**

This work proposes GraphMind, a framework that builds knowledge graph for LLMs to provide memory and help sequential decision making especially in BabyAI tasks.

**Strengths:**

The paper is clearly written and well structured. The use of a graph-based memory and DSL makes the system interpretable. Ablation studies (no memory, stacked memory) are clean and support the main claim. Graph-edit distance as a proxy for memory accuracy is intuitive.

**Weaknesses:**

I’m concerned about the generalization and scalability of the proposed framework. The experiments are confined to the BabyAI environment with only a few predefined layouts, which are relatively simple and small-scale. There’s no evaluation in more challenging or diverse settings, such as larger or irregular maze environments, tasks with richer object interactions, or real-world scenarios. So it’s unclear whether the approach would maintain its effectiveness beyond this narrow domain.

**Questions:**

Is the knowledge graph truly necessary? Couldn’t we achieve similar results by simply giving the LLM access to a global map of the environment or by providing prior trajectories and observations in plain text? Given the current strength of modern LLMs, such structured graph representations may not be essential for this relatively simple task.

---

> ### Author Response · Authors · 2025-11-25
>
> Thank you for your thoughtful comments. Please find the responses to your questions below.
>
> **Q1. Experiment on the BabyAI environment.**
>
> We focus on solving partially observable environments that provide only limited local information without any global information. BabyAI fits our purpose because it is a partially observable 2D gridworld that combines object manipulation, navigation, and mission execution. Due to the gridworld structure, the agent must plan precise trajectories around blockers, and the large 3×3 room configurations strongly require exploration under partial observability. We further increase the environment’s complexity by restricting visibility even after doors are opened.
>
> **Q2. Is the knowledge graph truly necessary?**
>
> Unlike textual or visual data, graph-structured data inherently encodes heterogeneous and relational information and supports queries that require a global understanding of the entire dataset ([1], [2]). In Table 2, our framework, which uses a knowledge graph as memory, demonstrates stronger performance than the linear, stack-structured memory (SM) in complex grid layouts and challenging missions.
>
> **Q3. Couldn’t we achieve similar results by simply giving the LLM access to a global map of the environment or by providing prior trajectories and observations in plain text?**
>
> While providing the LLM with a global map may achieve similar results, we assume that the agent receives only information limited by partial observability, such as objects within its current field of view, rather than global information like the exact locations of the agent or objects. This setting is more realistic for real-world scenarios, where a global map or precise localization is often unavailable (e.g., a robot navigating inside a building). Additionally, LLMs struggle to understand textual observations in context, as shown in [3]. Therefore, we focus on effective memory usage under this assumption, and our framework demonstrates stronger performance than simply providing stacked prior trajectories (SM in our experiments).
>
> [1] Han, Haoyu, et al. "Retrieval-augmented generation with graphs (graphrag)." *arXiv preprint arXiv:2501.00309* (2024).
>
> [2] Edge, Darren, et al. "From local to global: A graph rag approach to query-focused summarization." *arXiv preprint arXiv:2404.16130* (2024).
>
> [3] Paglieri, Davide, et al. "Balrog: Benchmarking agentic llm and vlm reasoning on games." *arXiv preprint arXiv:2411.13543* (2024).

---

### Note · Authors · 2026-01-02

I have read and agree with the venue's withdrawal policy on behalf of myself and my co-authors.